# Deleterious Mutations in the TPO Gene Associated with Familial Thyroid Follicular Cell Carcinoma in Dutch German Longhaired Pointers

**DOI:** 10.3390/genes12070997

**Published:** 2021-06-29

**Authors:** Yun Yu, Henk Bovenhuis, Zhou Wu, Kimberley Laport, Martien A. M. Groenen, Richard P. M. A. Crooijmans

**Affiliations:** Animal Breeding and Genomics, Wageningen University & Research, Droevendaalsesteeg 1, 6708 PB Wageningen, The Netherlands; yun.yu@wur.nl (Y.Y.); henk.bovenhuis@wur.nl (H.B.); Wuz1992@hotmail.com (Z.W.); Kimberley.laport@wur.nl (K.L.); martien.groenen@wur.nl (M.A.M.G.)

**Keywords:** dog, thyroid carcinoma, mutation, *TPO*, GWAS

## Abstract

Familial thyroid cancer originating from follicular cells accounts for 5–15% of all the thyroid carcinoma cases in humans. Previously, we described thyroid follicular cell carcinomas in a large number of the Dutch German longhaired pointers (GLPs) with a likely autosomal recessive inheritance pattern. Here, we investigated the genetic causes of the disease using a combined approach of genome-wide association study and runs of homozygosity (ROH) analysis based on 170k SNP array genotype data and whole-genome sequences. A region 0–5 Mb on chromosome 17 was identified to be associated with the disease. Whole-genome sequencing revealed many mutations fitting the recessive inheritance pattern in this region including two deleterious mutations in the TPO gene, chr17:800788G>A (686F>V) and chr17:805276C>T (845T>M). These two SNP were subsequently genotyped in 186 GLPs (59 affected and 127 unaffected) and confirmed to be highly associated with the disease. The recessive genotypes had higher relative risks of 16.94 and 16.64 compared to homozygous genotypes for the reference alleles, respectively. This study provides novel insight into the genetic causes leading to the familial thyroid follicular cell carcinoma, and we were able to develop a genetic test to screen susceptible dogs.

## 1. Introduction

In humans, thyroid cancer constitutes 3.4% of cancers diagnosed worldwide annually [1]. Thyroid carcinoma (TC) originating from follicular cells have two main types: follicular thyroid carcinoma (FTC) and papillary thyroid carcinoma (PTC). FTC accounts for 14% and PTC accounts for 81% [2] of thyroid carcinomas. In dogs, thyroid follicular cell carcinomas (FCC) are mainly classified into four types: FTC, PTC, compact thyroid carcinoma (CTC), and follicular-compact thyroid carcinoma (FCTC). These FCCs in dogs are remarkably similar in histology and biological behavior to thyroid carcinoma with follicular origin in humans [3]. Similarity in cell origin and histology of FCC indicates that dogs might be able to serve as a thyroid cancer model for research and treatments development.

Thyroid cancer can be of either familial or spontaneous origin, caused by heritable germline risk factor and sporadic somatic mutations, respectively. In humans, the genetics of TC were studied extensively. Genetic mutations are a major contributor to thyroid cancer [4]. Many germline genetic mutations were reported to be associated with familial TC, including mutations in *APC*, *PTEN*, *SDHB-D*, *PIK3CA*, *AKT1*, *SEC23B*, *WRN*, and *PRKAR1α*, which cause syndromic TC [5]. While most of these germline genetic mutations cause TC through a dominant mode. *WRN* gene mutations cause TC through an autosomal recessive mode [6]. Moreover, genome-wide association studies approaches identified many germline genetic mutations associated with familial TC. These include the genes *FOXE1* [7], *SRGAP1* [8], *HABP2* [9], *BRCA1* [10], *CHEK2* [11], *ATM* [12], *RASAL1* [13], *SRRM2* [14], *XRCC1* [15], and *PTCSC3* [16]. Most of these genes also cause TC through a dominant mode. Whole genome sequencing of thyroid tumor tissues identified many somatic mutations driving the initiation and the progression of TC. The type and the number of somatic mutations between cases with familial and spontaneous TC are similar [17]. *BRAF* (V600E) is the most common somatic mutation associated with PTC [18]. *RAS* somatic mutations are the second most common type of mutations found in fine needle aspiration of thyroid nodules [19]. Somatic *RAS* mutations are present in 15–30% of TC [20]. In addition, *PAX8/PPARG* was also identified to be an oncogenic driver for TC [19,21]. Somatic *RET*/*PTC* rearrangement associates with PTC [19]. *TERT* promoter somatic mutations were identified in follicular carcinoma specimens and may serve as a marker for the aggressive form of TC with lethal consequences [22,23]. TC in dogs can have the same genetic causes as TC in humans. For example, *K-RAS* somatic mutations were found in dogs with FTC and medullary thyroid carcinoma (MTC). Besides, different genetic causes were also identified. Germline mutations in the *RET* oncogene on chromosome 10q11.2 underlie most hereditary forms of MTC in humans with an autosomal dominant inheritance pattern [24], while, in dogs, the *RET* mutations were not found in hereditary MTC [25]. Compared to genetic research of TC in humans, research on the genetic background of TC in dogs is limited.

Familial thyroid carcinoma can be defined when two or more first-degree relatives are affected in the absence of other cancer predisposition syndromes [26]. Previously, we reported a large number of familial FCC in the Dutch German longhaired pointers (GLPs). The pedigree suggests a recessive mode of inheritance [27]. The aim of the current study was to identify the germline causal gene(s) of familial FCC in the GLP population. A variety of approaches was used, including a genome-wide association study (GWAS) and ROH analyses based on SNP array data. In addition, whole genome sequences of affected and unaffected GLPs were obtained to identify the potential causal/susceptible gene/variant in the candidate region, followed by validation of the candidate variants through PCR-RFLP in a larger number of Dutch GLPs. 

## 2. Materials and Methods

### 2.1. Animal and Diagnosis

All the German longhaired pointers used in this research were from the dataset described previously [27]. Briefly, in total, 264 GLPs were examined, and 84 cases were identified, of which 54 were histopathologically confirmed FCC cases, 1 was a follicular adenoma case, and 29 were suspected of thyroid neoplasia based on typical clinical signs such as the presence of cervical mass, but no further diagnostics were performed. Clinical examinations were performed by the veterinary oncology center “AniCural”. Blood of 186 GLPs and tumor samples 36 GLPs were collected by veterinarians at the time of the diagnosis. The owners of the dogs gave permission for the tissues to be used for research purposes. The histology assessments were performed by the Department of Pathology, Utrecht University. Detailed description of samples and diagnosis procedures can be found in the previous study [27].

### 2.2. Genotyping

DNA was extracted from animals genotyped and sequenced using Gentra Puregene Blood Kit (Qiagen, Hilden, Germany). Twenty-five affected and twenty-six unaffected GLPs were genotyped. The age at diagnosis of the genotyped dogs ranged between 4.5 and 9.8 years with an average of 7.3 years. All unaffected dogs had ages >13 years. Forty-three of the dogs were genotyped using the 170 k canineHD SNP beadchip array (Illumina Inc., San Diego, CA, USA). The remaining 8 dogs were genotyped using the 230 k canineHD SNP beadchip array (Illumina Inc., San Diego, CA, USA), which is the extended version of the 170 k canineHD SNP beadchip array (Illumina Inc., San Diego, CA, USA). Detailed information about the samples is in Appendix A. The 173,662 SNPs shared between these two SNP arrays were extracted for each genotyped dog and used in subsequent analyses. All the SNPs were remapped to the canine reference genome CanFam3.1 using the NCBI Remap tool. Quality control was performed using the following criteria: minor allele frequency > 0.05, maximum missing call rate per variant 0.1, and maximum missing genotype rate per individual 0.1. A total of 7398 variants were removed due to missing call rate.

To detect the genetic relationship between these GLPs genotyped, principal component analysis (PCA) was performed using Plink v1.9 [28], and the first two principal components were plotted using R. Inbreeding coefficient (F) based on the difference between observed and expected counts of autosomal homozygous genotypes was calculated using Plink v1.9 by --het command.

### 2.3. Whole Genome Sequencing (WGS) 

To identify the causal variant(s) in the candidate region identified by GWAS study, 22 GLPs (11 affected and 11 unaffected) were whole genome sequenced. DNA samples were used for library construction following the manufacture’s recommendations using NEB Next^®^ UltraTM DNA Library Prep Kit (Cat No. E7370L) (NEB, Ipswich, MA, USA). Index codes were added to each sample. Briefly, the genomic DNA was randomly fragmented to an average size of 350 bp. DNA fragments were end polished, A-tailed, ligated with adapters, size selected, and further PCR enriched. Then PCR products were purified (AMPure XP system, Beckman Coulter, Indianapolis, IN, USA), followed by size distribution by Agilent 2100 Bioanalyzer (Agilent Technologies, CA, USA) and quantification using real-time PCR. Libraries were sequenced on a NovaSeq 6000 S4 flow cell with PE150 strategy. Three dogs were sequenced at a depth of 30x 3 dogs at a depth of 60x, and 16 dogs at a depth of 10x. All the healthy dogs had an age above 13 years by the time of the study. Detailed information about the samples can be found in Appendix A. FastQC [29] was used to evaluate the quality of the sequences. Sickle was used to trim the reads using default settings. Sequences were aligned to the CanFam 3.1 reference genome using BWA-MEM algorithm (version 0.7.15) [30], then samtools 1.9 [31] was used to sort the aligned reads and to remove duplications. GATK3.5 [32] was used to perform indel-based re-alignment. 

Freebayes [33] was used to call single-nucleotide variants (SNPs) and small insertions or deletions (InDels) from WGS for each dog. Filtering was performed using bcftools v1.9 [34]. Loci covered by less than 4 reads were removed. In addition, variants with a calling quality less than 20 were also discarded. Structural variants were called using Manta [35]. Nine WGSs (GLP77, GLP44, GLP39, GLP84, GLP85, GLP169, GLP82, GLP04, GLP25) were used for SV calling. The other WGSs were discarded due to uneven coverage across the genome. Variants were annotated and analyzed for predicted effects using VEP [36] program and were visually confirmed in Jbrowse [37].

### 2.4. RNA-Sequencing and Data Processing

Tumor tissues of left thyroid gland from 7 affected dogs were sampled at the time of diagnosis and stored in RNAlater RNA stabilization reagent (Qiagen, Hilden, Germany). RNA was extracted from the tumor tissue using AllPrep RNA Mini Kit (Qiagen, Hilden, Germany) according to manufacturer’s instructions. The RNA samples were used for library preparation. The directional libraries were prepared using NEBNext^®^ Ultra TM Directional RNA Library Prep Kit for Illumina^®^ (NEB, Ipswich, MA, USA) following manufacturer’s protocol. Indices were included to multiplex multiple samples. Briefly, mRNA was purified from total RNA using poly-T oligo-attached magnetic beads. After fragmentation, the first strand cDNA was synthesized using random hexamer primers followed by the second strand cDNA synthesis. The strand-specific library was ready after end repair, A-tailing, adapter ligation, size selection, and USER enzyme digestion. After amplification and purification, insert size of the library was validated on an Agilent 2100 and quantified using quantitative PCR (Q-PCR). Libraries were then sequenced on the Illumina NovaSeq 6000 S4 flowcell with PE150 according to results from library quality control and expected data volume. FastQC were used to check the read quality. Hisat2 [38] was used to map the reads to reference genome CanFam3.1 with --dta option. Then, FeatureCounts [39] was used to quantify mapped reads to genomic features such as genes, exons, gene bodies, genomic bins, and chromosomal locations. Alignments were visually inspected in IGV [40].

### 2.5. GWAS

GEMMA 0.98.1 [41] was used to perform the genome wide association analysis using an univariate linear-mixed model, correcting for population stratification by accounting for family relationships among dogs by incorporating a standardized genomic relationship matrix, calculated from SNP array data by GEMMA. There were 45,819 SNPs discarded from the analysis by the default filtering of GEMMA. Manhattan plots were generated by plotting *p*-value of Wald test using qqman package in R. 

### 2.6. Runs of Homozygosity 

Previous analyses suggest a recessive mode of inheritance. In that case, affected dogs are expected to carry two copies of the causal allele. Therefore, we expected, in the region carrying the causal mutation, a run of homozygosity (ROH) in affected dogs while expecting it would be absent in unaffected dogs. Runs of homozygosity across the genome were detected using PLINK v1.9. ROHs were defined according to the following criteria: (i) the minimum count of SNPs in a sliding window was 15; (ii) the minimum ROH length was set to 1 Mb; (iii) the maximum inverse density was 100 Kb per SNP; (iv) to avoid the effects of low SNP density region, the maximum gap length between consecutive SNPs was 1 Mb; (v) the minimum hit rate of all scanning windows containing the SNP was set to 0.05. The ROH autozygosity on each chromosome was plotted contrasting affected and unaffected dogs using in-house R script.

### 2.7. Candidate SNPs PCR-RFLP Genotyping

PCR assay was done using 60 ng of genomic DNA with 0.4 µM of each primer and 5 × FIREPol^®^ Master Mix, 7.5 mM MgCl2 (Solis BioDyne, Estonia) in a final volume of 12 µL. PCR primers for chr17:800788G>A were Forward 5′- CAGGTTACAACGCGTGGAG -3′ and Reverse 5′- TCCCTCAGAGCCTTCATCTG -3′ to generate a 232 bp amplicon. The PCR primers for chr17:805276C>T were Forward 5′- AGGGTGGTTTCAGGTGTGAG -3′ and Reverse 5′- GTGAGGACACGGCAAGAGAT -3′ to generate a 172 bp amplicon. The PCR reaction was carried out in a T100 Thermal Cycler (BioRad, CA, USA) and included an initial denaturation for 1 min at 95 °C was followed by 35 cycles of 95 °C for 30 s, 55 °C for 45 s, and 72 °C for 90 s, followed by a 5 min extension at 72 °C. The electrophoresis of PCR products was performed in 1.5% agarose gel containing Stain G (Serva, Germany) together with a 100 bp DNA ladder (New England Biolabs, Ipswich, MA, USA) and photographed using a gel documentation imaging system (BioRad, Hercules, CA, USA). Regarding the RFLP of *TPO* gene PCR product, the 232 base pair product of chr17:800788G>A was digested with BssHII restriction enzyme (New England Biolabs, Ipswich, MA, USA) according to manufacturer’s instructions to generate fragments (5 h at 50 °C followed by 20 min at 65 °C). Digestions were carried out in a total volume of 10 μL. The reaction mixture consisted of 5 μL of PCR product, 5 U of restriction enzyme/Cutsmart Buffer, and volume adjusted with sterile distilled water. The 172 base pair product of chr17:805276C>T was digested with Hpy99I restriction enzyme (New England Biolabs) according to manufacturer’s instructions to generate fragments (5 h at 37 °C followed by 20 min at 65 °C). Digestions were carried out in a total volume of 10 μL. The reaction mixture consisted of 5 μL of PCR product, 2 U of restriction enzyme/Cutsmart Buffer, and volume adjusted with sterile distilled water. The digest was electrophoresed in 3% agarose with Stain G (Serva, Germany) together with a 100 bp DNA ladder (New England Biolabs, Ipswich, MA, USA) and photographed using a gel documentation imaging system (BioRad, Hercules, CA, USA).

### 2.8. Criteria for Candidate Variants

We called variants in the candidate region using whole genome sequencing. The genotypes of the candidate variants associated with the familial FCC in affected and unaffected dogs were to fit the following pattern: affected dogs (excluding GLP04 and GLP60, reason is shown in the result Section 3.2) should be homozygous, and all unaffected dogs should be heterozygous or homozygous for the alternative allele; the alternative allele frequency in NCBI dbSNP should be lower than 0.05. Additionally, for the mutations in the exonic region, we focused on the mutations predicted to be deleterious by SIFT [42], PROVEAN [43], PANTHER-PSEP [44], and PolyPhen-2 [45].

### 2.9. Amino Acid Conservation between Species

TPO amino acid sequences of six species (Human, Dog, Pig, Chicken, Mouse, Rhesus macaque) were obtained from NCBI and aligned using Clustal Omega from EMBL-EBI.

## 3. Results

### 3.1. Study Population

From the affected GLPs we previously described, 25 FCC affected GLPs were selected for SNP array genotyping. Affection status of 23 GLPs was confirmed by histology, while 2 cases were suspected based on typical clinical signs, e.g., the cervical mass (Appendix A). Of the 22 dogs that were sequenced, 11 were affected, of which 3 were suspected based on clinical signs (Appendix A).

The GLPs were highly inbred with a mean inbreeding coefficient estimated based on difference between observed and expected homozygotes of 0.50. Inbreeding was higher in affected (0.51) than in unaffected dogs (0.48) (Welch two sample test: *p*-value of 0.002) (Figure 1a), which is in agreement with our previous analysis based on inbreeding coefficients estimated from the pedigree [27]. In the genotyped affected dogs, 72% were male (18 out of 25), and in the unaffected dogs, this was 38% (10 out of 26).

As reported earlier [27], the FCC in these Dutch GLPs is a heritable disease and very likely a recessive trait. Most affected dogs are closely related. Here, the principal component analysis shows that affected dogs were clearly separated from unaffected dogs (Figure 1b). This emphasizes the importance of accounting for family relations in the association analysis in order to control for false positive discoveries.

### 3.2. Genomic Region Associated with FCC

To identify the genomic region responsible for FCC, a combination of two different methods was used: a GWAS and an ROH analysis. The GWAS analysis identified the region associated with the disease. The ROH analysis identified the homozygous genomic region present in the affected dogs, while it was absent from the unaffected dogs. To increase the statistic power of the GWAS analysis, we combined SNP array genotype data and WGS data to obtain a larger sample size for a total of 103744 SNPs shared by the two methods. Three whole-genome sequenced cases (GLP25, GLP60, GLP77) and 10 controls (GLP84, GLP85, GLP115, GLP124, GLP160, GLP168, GLP169, GLP170, GLP171, GLP172) were added to the panel to achieve a sample size of 64 (28 cases, 36 controls). The other GLPs were discarded from the analysis because of either a late age at diagnosis (>10 years) or being already genotyped with the SNP array. The signal on chromosome 17 was captured by GWAS. The Manhattan plot of the GWAS result (Appendix A) across the whole genome and the qq plot (Appendix A) is shown in the Appendix A. Furthermore, a long ROH segment (Figure 1c) overlapping the location of the signal of GWAS was present in the affected dogs, while it was absent from the unaffected dogs. The autozygosity of ROH across the whole chromosome 17 is shown in Appendix A and autozygosity on each other chromosome is shown in Appendix A. Starting from the position chr17:4741065, the autozygosity in affected dogs dropped from 86% to 71% and continued to drop further, while, in unaffected dogs, the autozygosity remained low at 8% to 14%. Therefore, we set the region 0–5 Mb as the candidate region, which was somewhat wider and covered the homozygous region. This region was also supported by the haplotypes of dogs with WGS data (Figure 1d), where the long ROH segment broke at a position close to 5 Mb (indicated by a red arrow in Figure 1d), as two dogs (GLP44 and GLP36) appeared to be heterozygous from thereon. In the candidate region, five SNPs surpassed the Bonferroni corrected significance threshold (–log10(*p*) = 6.3) in the GWAS analysis. There were 13 known protein coding genes in the region: *SNTG2*, *TPO*, *PXDN*, *MYT1L*, *EIPR1*, *TRAPPC12*, *RPS7*, *RNASEH1*, *COLEC11*, *DCDC2C*, *ALLC*, *RSAD2*, and *RNF144A*.

In the candidate region, 2 of 11 sequenced affected GLPs (GLP04 and GLP60) showed distinct haplotypes from the other 9 affected GLPs (Figure 1d). GLP04 has a very late onset age (13.5 years) and could be a spontaneous case with different genetic or environmental causes. Another dog, GLP60, was a suspected case without histology confirmation. It could be actually affected by other thyroid diseases which show similar clinical signs rather than FCC. Therefore, these two GLPs were excluded from cases when using WGS to identify case-specific variants.

The WGS analysis revealed 23,338 variants (SNPs and InDels) in this 5 Mb candidate region. Among them, 2374 variants were case-specific (excluding GLP04 and GLP60) intronic, intergenic, synonymous, and nonsynonymous homozygous mutations (SNP and InDel) (Table 1) in the region of 0–5 Mb chr17. Three mutations (two in the *TPO* gene, one in the *SNTG2* gene) were predicted to be deleterious by in silico pathogenic prediction tools. SVs were also called in the candidate region, and case-specific SVs are shown in Table 2. RNA-seq of FCC tumor from seven affected GLPs was used to check the mRNA expression and architecture by visual inspection in IGV. No SV was found to change the mRNA structure of the corresponding genes according to the inspection of the mRNA expression. They were excluded as candidate causal variants.

### 3.3. Deleterious Mutations in the TPO Gene

In the *TPO* gene, two missense mutations, chr17:800788G>A (686F>V) and chr17:805276C>T (845T>M), were identified in the WGS data from 22 GLPs (11 affected and 11 unaffected) which were exclusively found in affected dogs. These mutations were predicted to be deleterious by several pathogenic prediction tools (SIFT, PROVEAN, PANTHER, PolyPhen-2) (Table 3). Variant chr17:800788G>A (686F>V) is not present in 722 canine genomes [46] from over 144 modern breeds, 54 wild canids and 100 village dogs. It is a novel mutation that is not yet annotated in NCBI dbSNP. Variant chr17:805276C>T (845T>M) was detected at a very low allele frequency of 2%, with only 6 homozygotes and 15 heterozygotes in the 722 dogs. In our sequenced GLPs, 9 of the 11 affected dogs were homozygous for both variants. The other two exceptions were GLP04 (homozygous for reference allele) and GLP60 (heterozygous). No unaffected dogs were homozygous for the two variants (nine heterozygotes and two homozygotes for the reference allele). The genotypes for the two sites fit the autosomal recessive inheritance pattern. 

Structural variants were also noticed but no structural variants were found in the *TPO* gene. According to the Sashimi plot from RNA of tumor tissues of seven affected dogs in the IGV (Appendix A), the mRNA structure of the gene did not change without alternative splicing events or gene fusion.

The *TPO* gene encodes an enzyme named thyroid peroxidase, which is a poorly glycosylated membrane-bound enzyme. It is involved in thyroid hormone synthesis and a target autoantigen in autoimmune thyroid disorders. TPO oxidizes iodide ions to form iodine atoms for addition onto tyrosine residues on thyroglobulin for the production of thyroxine (T4) or triiodothyronine (T3), the thyroid hormones [47].

Both variant locations are conserved across species (Figure 2). Canine TPO c.F686 corresponds to human TPO h.678 located in the MPO-like domain of the protein. The MPO-like domain consists of two immunodominant regions. Canine TPO c.T845 corresponds to human TPO h.837 located in a conserved calcium-binding EGF-like domain. The EGF-like domain is involved in ligand recognition and protein–protein interaction. The amino acid changes in the region may change the three-dimensional structure, which could impact the catalytic activity or the autoimmunity of TPO [48].

Except for the 2 deleterious mutations, 4 synonymous mutations and 31 intronic variants were also identified in the *TPO* gene of affected dogs. These variants were in strong LD with the two deleterious mutations and were less likely to be the causal mutations and therefore were not investigated further.

### 3.4. Deleterious Mutation in the SNTG2 Gene

In the *SNTG2* gene, the WGS analysis revealed a case-specific variant 743943T>C (360F>S) (Table 3), which is also a rare mutation with an alternative allele frequency of 0.02 in the 722 dogs [46]. This variant was predicted to be deleterious by SIFT, while it was predicted to be a neutral mutation by PROVEAN, probably benign by PANTHER, and benign by PolyPhen-2. Similar as for the two deleterious mutations found in the *TPO* gene, 9 of 11 affected dogs were homozygous for this mutation, while GLP60 was heterozygous and GLP04 was homozygous for the reference allele. None of the unaffected dogs were homozygous for this mutation.

### 3.5. Variants in the EIPR1 Gene

The *EIPR1* gene (also named *TSSC1*, tumor-suppressing subchromosomal transferable fragment candidate gene 1) codes for a protein that acts as a specific interactor of both GARP (Golgi-associated retrograde protein) and EARP (endosome-associated recycling protein), playing a critical role in endosomal retrieval pathways [49]. The *EIPR1* gene harbors 10 variants that fit a recessive inheritance pattern (Table 4). All the cases, including GLP04 and GLP60, were homozygous for these mutations, while controls were heterozygous or homozygous for the reference alleles. All 10 mutations were located in introns or the downstream region of the gene. None of these variants were predicted to affect the mRNA structure of the gene. Additionally, within the 722 dogs, homozygotes for the alternative alleles at these loci were relatively common (Table 4). Therefore, these mutations were unlikely to play a critical role in thyroid tumor development and were excluded as potential candidate causal variants for FCC in this study.

### 3.6. Validation by PCR-RFLP

To confirm the association between the two mutations in the *TPO* gene and the familial FCC, we genotyped chr17:800788G>A in a further 59 cases and 123 controls and chr17:805276C>T in 59 cases and 127 controls using PCR-RFLP (Table 5). Genotype of each dog can be found in Appendix A. For both variants, 45 of 59 cases (76%) were homozygous and 9 (7%) and 10 (8%) controls were homozygous for the two variants, respectively. Totals of 83% and 82% of dogs with homozygous variant for the two mutations were affected, respectively. These totals suggest that these two mutations in the *TPO* gene were highly associated with the FCC with a *p*-value < 2.2 × 10^−16^ for the Fisher’s exact test. Furthermore, chr17:800788G>A had lower SIFT and PROVEAN scores than chr17:805276C>T, and it is a novel mutation. Therefore, chr17:800788G>A was of more interest than chr17:805276C>T. Homozygous mutants of both variants had extremely high relative risk (16.94 and 16.64, respectively) compared to the homozygous genotype for the reference alleles. The heterozygous genotypes had a higher relative risk, but the differences were not significant. According to affection status and genotypes of the dogs in the pedigree in Figure 3, along with the long ROH segment in affected dogs, these two mutations were associated with the FCC in dogs in an autosomal recessive inheritance pattern. Among those 14 non-homozygous cases, 5 dogs were suspected cases without histology confirmation, which could be mis-diagnosed. In the remaining 9 cases with histology diagnosis, 6 dogs had ages at diagnosis beyond 10 years. One dog had unknown age at diagnosis. Only 2 dogs had age at diagnosis less than 10 years. The affected dogs with old age at diagnosis (we used a threshold of 10 years in this study) possibly had a somatic genetic causal mutation due to an environmental risk factor or aging. Ten of 124 unaffected dogs were homozygous. Among them, seven dogs were born after 2007 (four dogs after 2012). They were fewer than 12 years old at the moment of the data collection and could be affected at an older age. 

## 4. Discussion

Many dog breeds experienced considerable inbreeding and show comparable diversity loss compared to other domestic species due to artificial selection, management in closed populations, and historical bottlenecks [50]. Inbreeding depression in the form of a variety of diseases and disorders is seen in many dog breeds due to this loss of genetic diversity [51]. Based on pedigree data, we previously showed that affected dogs are more inbred than unaffected dogs [27]. This was confirmed by the inbreeding coefficients estimated from SNP array genotype data. Likewise, affected dogs have more ROH segments above 2 Mb in length (Appendix A), which also implies that affected dogs exhibit more inbreeding.

We identified a region located between positions 0 and 5 Mb on chromosome 17 that is associated with FCC in the Dutch GLPs using a combination of GWAS and ROH analyses. In the affected dogs, this region showed a loss of diversity. The causal mutation located in a long ROH segment resulting from inbreeding was captured by ROH autozygosity analysis.

Using whole genome sequencing, we identified three rare deleterious mutations of interest within *SNTG2* and *TPO* genes, located in the long ROH region on chr17. Mutation Chr17:800788G>A, located in the *TPO* gene, was never reported. The other two have very low allele frequency in dogs from a variety of breeds. *SNTG2* is a syntrophin gene. The SNTG2 protein binds to components of mechanosensitive sodium channels and to the C termini of dystrophin, α-dystrobrevin, and β-dystrobrevin [52]. The *SNTG2* gene is expressed in various tissues in humans and was reported to be associated with osteoporotic vertebral fracture [53] and autism [54]. These give the gene an unlikely role in the development of TC. Additionally, no common structure change of *SNTG2* mRNA was found within the cases from the RNA-seq (Appendix A). The deleterious mutations in the *TPO* gene are highly associated with the familial FCC in these dogs. However, the high linkage (5 Mb) in this study opens the possibility that other mutations in the noncoding regions could also explain the association. Nonetheless, *TPO* mutation chr17:800788G>A is of more interest with the lowest SIFT and PROVEAN scores and the aspect of it being a novel mutation. 

The association between *TPO* gene mutations including intronic variants and missense variants and thyroid carcinoma was also seen in humans [55,56]. Mutations in the *TPO* gene also cause congenital goitrous primary hypothyroidism. Inactivating mutations in *TPO* gene were shown to cause the autosomal recessive trait congenital hypothyroidism in humans and dogs [57]. 

*TPO* is expressed specifically in thyroid gland tissue. Germline genetic alterations in other thyroid-specific genes were also associated with thyroid carcinoma. In humans, the rs965513[A] allele, which confers the greatest relative risk for the development of thyroid cancer, is located within an enhancer element controlling expression of *FOXE1*. *FOXE1* is a thyroid-specific transcription factor that regulates several genes involved in thyroid hormone production, including *TPO*, thyroglobulin (*TG*), sodium-iodide symporter (*SLC5A5*), and dual oxidase (*DUOX2*). Furthermore, *TG* [58], *FOXE1* [59], and *DUOX2* [60] were also shown to be associated with thyroid cancer. Mutations in *SLC5A5* are associated with congenital hypothyroidism [61].

Although the *TPO* gene was identified to be associated with thyroid cancer and many other thyroid disorders, how TPO influences the risk of TC is still unclear. Up to 70% of TCs are caused by somatic mutations that activate the RAS/ERK mitogenic signaling pathway (MAPK/ERK) [21]. Upregulation of mitogen-activated protein kinase (MAPK) and phosphatidylinositol-3-kinase (PI3K)/Akt signaling pathways was reported to cause the thyroid gland tumorigenesis in dogs and humans [3]. Here, in dogs, the mechanism through which we identified deleterious mutations in the *TPO* gene influence risk of FCC may be more similar to that of the mutations associated with TC found in other thyroid-specific genes, for example, dysregulated hydrogen peroxide metabolism [60], because these genes work closely together to synthesize thyroid hormones. 

We sequenced the RNA derived from the tumor tissue of seven affected dogs. The mRNA of the *TPO* gene was not interrupted (Appendix A). However, it is not known whether the *TPO* mRNA expression in the tumor changed compared to the expression in normal canine thyroid gland. Likewise, this is not known for other genes in the long ROH within this region. However, mutations in the regulatory region, e.g., promoter and enhancer, could alter the expression level of the gene and thereby induce the tumor. Thus, further studies focusing on gene expression differences between affected and unaffected dogs are needed. Moreover, non-coding RNA, including lncRNA, microRNA, and circRNA, were shown to be involved in the tumorigenesis of thyroid tumor [1]. The lack of RNA-seq data from the normal thyroid tissue prohibited investigating the expression of the non-coding RNA genes.

Animal models for specific human diseases can contribute to research and treatment development. Many mouse models for thyroid cancer with varied genetic causes regarding different types of thyroid cancer were induced [62]. However, skepticism about their relevance with human thyroid cancer and their value in clinical translation is also presented mainly due to the difference between mice and humans and the fact that there is a very low translational rate of approximately 4–5% [63]. Dog models for some diseases were already introduced. For instance, a dog model for Alzheimer’s disease was generated by overexpressing a mutated human amyloid precursor protein [64]. Likewise, a canine model of glycogen storage disease type Ia (GSDIa) was also described with causal mutations in the same gene [65]. Compared to rodent models, dog models have many advantages. For instance, dogs are more similar to human in genetics, physiology, and living environment compared to rodents. Dogs receive good medical care, especially in developed countries; therefore, diseases in dogs are easily identified. Many dog breeds are predisposed to specific diseases, which can provide sufficient numbers of naturally affected dogs for research. Dogs can also benefit from research and treatment development using dog models. The treatment successfully developed from the model can also cure the disease in dogs.

## 5. Conclusions

In conclusion, we identified two deleterious recessive mutations in the *TPO* gene which are highly associated with the familial FCC in the Dutch GLPs. These findings provide a novel candidate gene and new insights to the tumorigenesis of FCC. A genetic test can be developed for veterinary diagnostic and selective breeding to eradicate this disease from the population. The dogs closely related to the affected dogs diagnosed are valuable to serve as a disease model for research or treatment development of TC caused by the alteration in genes involved in the thyroid function molecular pathway.

## Figures and Tables

**Figure 1 genes-12-00997-f001:**
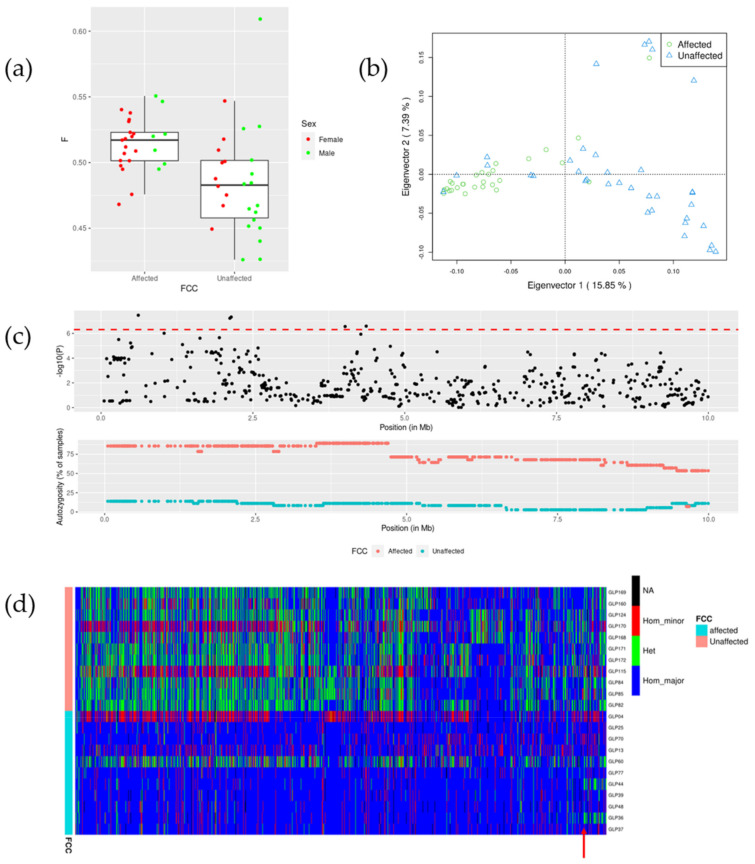
(**a**) Inbreeding coefficient of affected and unaffected dogs based on SNP chip genotype data. (**b**) PCA plot. First and second components are plotted for 28 affected and 36 unaffected GLPs used in the GWAS analysis. Most affected dogs are clustered apart from unaffected dogs, indicating clear population stratification. (**c**) Manhattan plot and autozygosity of ROH segments in affected and unaffected GLPs in the region between 1–10 Mb on chr17. The red dashed line in the Manhattan plot denotes the Bonferroni corrected significance threshold. (**d**) Genotypes of the variants between 0 and 5 Mb on chromosome 17 identified by WGS from 22 dogs. Colors blue, green, red, and black denote homozygous for major allele, heterozygous, homozygous for minor allele, and missing genotype, respectively.

**Figure 2 genes-12-00997-f002:**
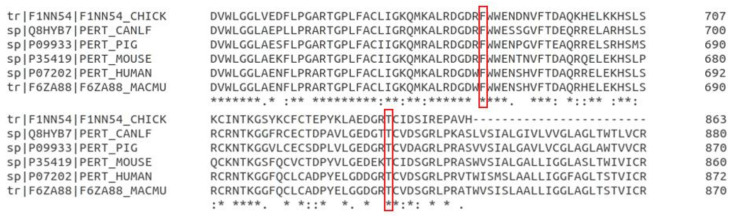
Conservation of the two amino acids of the TPO corresponding to the two deleterious mutations between six species. The two deleterious mutation loci (indicated in red box) are very conserved across species.

**Figure 3 genes-12-00997-f003:**
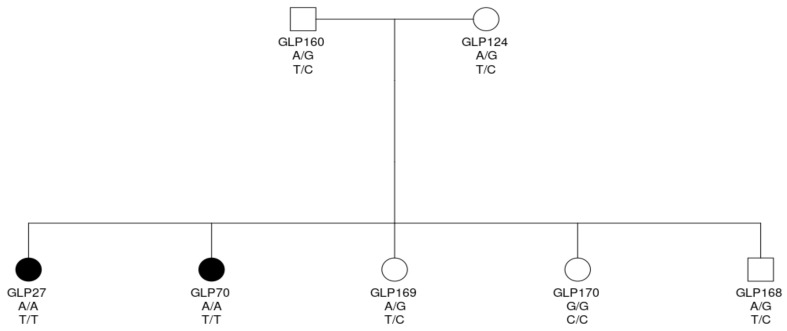
Genotypes of dogs suggesting a recessive trait of the disease. A circle denotes a female dog and a square denotes a male dog. Black background indicates that the dog is affected.

**Table 1 genes-12-00997-t001:** Variants identified via the whole genome sequence of GLPs.

Total Number of Variants	12,248,323
Variants in the candidate region	23,338
Of which are homozygous	6171
Of which are private for cases	2374
Of which are exonic	18
Of which are missense	7
Of which are deleterious	3

**Table 2 genes-12-00997-t002:** Private SV variants for cases.

Chromosome Coordination	SV-Type	Location	SV-Length
chr17:370832-370908	deletion	intergenic	77
chr17:379867-380195	deletion	intergenic	328
chr17:491919-492126	deletion	intergenic	208
chr17:503763-503968	deletion	intergenic	206
chr17:533005-533231	deletion	intergenic	227
chr17:551908-551961	deletion	intergenic	54
chr17:626018	insertion	intergenic	95
chr17:731556	insertion	Intron-*SNTG2*	55
chr17:885408-885477	deletion	downstream	70
chr17:900381-900486	deletion	intergenic	106
chr17:1633933	insertion	intergenic	57
chr17:1859631	insertion	intergenic	214
chr17:1938475-1938524	deletion	Intron-*EIPR1*	50
chr17:2136862-2137405	deletion	upstream	544

**Table 3 genes-12-00997-t003:** Genotype counts of candidate SNPs.

Genomic Coordinates	Gene	Amino Acid Change	SIFT	PROVEAN SCORE (Cutoff = −0.25)	PANTHER	PolyPhen-2	Genotype Counts ^1^		
							Affected GLPs	Unaffected GLPs	722 Dogs ^2^
Chr17:800788G>A	*TPO*	686F>V	Deleterious (0)	−6.775	0.89	0.999	9/1/1	0/9/2	-
Chr17:805276C>T	*TPO*	845T>M	Tolerated (0.06)	−4.042	0.89	1	9/1/1	0/9/2	6/15/637
Chr17:743943T>C	*SNTG2*	360F>S	Deleterious (0)	−1.901	0.27	0.337	9/1/1	0/9/2	4/16/615

Note: ^1^ counts of recessive homozygotes/heterozygotes/homozygotes for the reference allele. ^2^ 722 dogs covering 144 modern breeds, 54 wild canids and 100 village dogs.

**Table 4 genes-12-00997-t004:** Case-specific SNPs found in the EIPR1 gene.

Chromosome	Position	Gene	Element	Genotype Counts ^1^		
				Case	Control	722 Dogs ^2^
17	1869833	*EIPR1*	Downstream gene	11/0/0	0/10/1	38/93/536
17	1898881	*EIPR1*	4th intron	11/0/0	0/10/1	169/190/318
17	1898831	*EIPR1*	4th intron	11/0/0	0/10/1	37/117/528
17	1898919	*EIPR1*	4th intron	11/0/0	0/9/2	163/190/326
17	1901542	*EIPR1*	4th intron	11/0/0	0/10/1	112/170/376
17	1901551	*EIPR1*	4th intron	11/0/0	0/10/1	102/166/384
17	1901564	*EIPR1*	4th intron	11/0/0	0/10/1	104/172/373
17	1905133	*EIPR1*	4th intron	11/0/0	0/10/1	182/214/315
17	1933629	*EIPR1*	3rd intron	11/0/0	0/10/1	174/222/315
17	1948595	*EIPR1*	3rd intron	11/0/0	0/10/1	161/182/338

Note: ^1^ count of recessive homozygotes, heterozygotes, and homozygotes for the reference allele. ^2^ 722 dogs from over 144 modern breeds, 54 wild canids and 100 village dogs.

**Table 5 genes-12-00997-t005:** Genotypes of the two deleterious SNPs in the *TPO* gene in the Dutch GLPs.

SNP	AA	AR	RR	Relative Risk of AA	Relative Risk of AR
chr17:800788G>A	45/9	11/56	3/58	16.94 (*p*-value 2.20 × 10^−16^)	3.34 (*p*-value 0.07)
chr17:805276C>T	45/10	11/59	3/58	16.64 (*p*-value 2.24 × 10^−16^)	3.20 (*p*-value 0.09)

Note: A represents alternative allele, and R represents reference allele. Numbers in the cell are the counts of affected and unaffected GLPs. The *p*-value was derived from chi-square test.

## Data Availability

Sequencing data presented in this study are openly available at EMBI-EBL ENA database with reference number PRJEB43017. SNP array genotype data are available through ArrayExpress (accession number E-MTAB-10241).

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
