# Peer review of "Deleterious Mutations in the TPO Gene Associated with Familial Thyroid Follicular Cell Carcinoma in Dutch German Longhaired Pointers"

_genes, 2021, doi:10.3390/genes12070997_

Round 1
Reviewer 1 Report
"Recessive deleterious mutations in the TPO gene associated 2 with familial thyroid follicular cell carcinoma in Dutch German longhaired pointers" is an original article indagating mutations in the TPO gene in thyroid carcinoma in pointers. The work is well designed and well presented. The Authors identified two deleterious recessive mutations in the TPO gene, which is an interesting and original finding.
Author Response
Dear reviewer,
Thanks to the comments on the manuscript. According to the comments, no changes to the manuscript are needed.
Yours sincerely
Reviewer 2 Report
in this manuscript, the authors reported the identification of recessive mutations in the TPO gene in Dutch German longhaired pointers affected by familiar thyroid follicular cell carcinoma. I think that the manuscript is very interesting and well written. Also the methodologies are appropriate and the results are well reported. I have only some points to be addressed:
- Do you know the clinical phenotype of the reported cases of dogs carrying the already known variants in TPO and SNTG2 genes?
- In addition to SIFT and PROVEAN, I should suggest the authors to use other in silico prediction tools in order to better confirm the pathogenicity of the identified variants.
Author Response
Dear reviewer,
Thanks for the comments on our manuscript. We carefully addressed the comments raised in your review. Our response to each comment is listed in bold below. The changes to the manuscript were also indicated.
- Do you know the clinical phenotype of the reported cases of dogs carrying the already known variants in TPO and SNTG2 genes?
Response: We have the age at diagnosis, T4 at diagnosis, TSH at diagnosis for some affected dogs with genotype of the variants found in TPO and SNTG2 gene. But not every affected dog were tested on T4 and TSH level. These information were provided in supplementary table S2. Additionally, because the variant in SNTG2 gene was not genotyped in the whole Dutch GLPs we have. Therefore, only 6 dogs have the genotype information of SNTG2 variant that was obtained from WGS, as you can find in the table S2. Meanwhile, we also updated the supplementary table S1. Genotypes for the two candidate causal variants of each dog were included in that file.
T4 level is available for 20 homozygous cases (11 normal, 8 low, 1 high) and TSH level is available for 4 homozygous cases (1 normal, 1 low, 2 high). Sixteen of the 20 homozygous case have bilateral tumours. There are some other clinical signs recorded for some affected dogs as well. Detection of palpable thyroid mass without any other concurrent signs was reported in the majority of dogs (37). Seven dogs demonstrated additional clinical signs that included: intermittent cough (3 dogs), alopecia (3 dogs), polyuria (2 dogs) polydipsia (2 dogs), weight loss (1 dog), and lethargy (1 dog).
- In addition to SIFT and PROVEAN, I should suggest the authors to use other in silicoprediction tools in order to better confirm the pathogenicity of the identified variants.
Response: As you suggested, we used two other prediction tools, PolyPhen-2 and PANTHER-PSEP, which also confirmed that the two mutations in the TPO gene might be deleterious. These prediction results were added to the Table 3 of the manuscript.
|
Substitution |
Pdel (PANTHER) |
PolyPhen-2 |
|
F686V (Chr17:800788G>) |
0.89 (Probably damaging) |
0.999 (Probably damaging) |
|
T845M (Chr17:805276C>T) |
0.89 (Probably damaging) |
1 (Probably damaging) |
Reviewer 3 Report
Yu et al. studied a cohort of German Longhaired Pointers with a very high incidence of thyroid carcioma (TC). In a previous preprint (ref. 14.), the same group reported clinical and histopathological details as well as an estimated heritability of 0.62. In ref. 14, the authors also reported (weak) evidence that a major risk allele might have an autosomal recessive mode of inheritance. The data from this preprint is essential for the present manuscript. It is unfortunate that the preprint has not yet successfully passed a normal peer review process.
The familial TC in this cohort represents an extremely valuable and possibly unique resource to identify a germline risk factor for TC and thus to increase our mechanistic understanding of thyroid cancer. In the present manuscript, Yu et al. attempted to identify the causative genetic variant(s) for this TC risk factor. The topic of this investigation is of high interest to a broad audience of scientists. Unfortunately, the experimental design of the study is inappropriate and the claims of the authors are not sufficiently supported by the available data.
I hope that my comments will help the authors to perform more meaningful experiments and analyses that will allow a better utilization of their valuable samples.
Specific comments:
(A) Fundamental comments:
A1.
The mapping of the risk locus does not conform to currently accepted standards and must be considered inconclusive. I think that the risk locus on chromosome 17 might be real, but the presented data do not sufficiently support this claim.
A2.
The identification of the candidate causative variant is not plausible. The claims of "causative variants" are not sufficiently backed by the data and I am firmly convinced that none of the specifically reported variants has any functional relevance for the cancer risk in this cohort.
(B) Major comments:
B1.
The risk locus was mapped by 3 complementary approaches. I see problems with each of these experiments. I start with the GWAS: The GWAS did not yield any genome-wide significant associations (Figure S2). The QQ-plot also does not indicate any significant deviation of the observed from the expected p-values. As the authors have access to more samples, they should perform the GWAS with additional samples to increase the statistical power. The authors alternatively might consider to perform linkage analyses as they have access to complete families.
B2.
Selective sweep analysis: This approach is not appropriate to map a cancer susceptibility locus. Developing a cancer will hardly provide a selective advantage.
B3.
Homozygosity analysis: If the risk allele is truly recessive, homozygosity analysis may actually represent the best approach to obtain a precise fine-mapping of the critical interval. If the authors are convinced that this familial cancer syndrome can be seen as a (nearly) monogenic trait with recessive inheritance, then all cases must share a common haplotype segment in homozygous state. To me, it is acceptable to retrospectively exclude a small number of cases that don't share the same haplotype, if there are plausible and consistent phenotypic criteria that justify such an exclusion (e.g. an unusually late age of onset). The homozygosity will also become more meaningful and more accurate, if all available cases are included. A precisely defined critical interval should result from the homozygosity analysis (with bp coordinates that should be explicitly stated in the manuscript) .
B4.
The description of the samples is sloppy and inconsistent. In the methods, it is stated "All the German longhaired pointers used in this research were from the dataset described previously [14]. Briefly, in total, 264 GLPs were examined and 84 cases were identified, ..." In ref. 14, only 54 cases are mentioned! Table S1 should be expanded, so that it contains all available data on each dog in the study. This table should list all phenotype data and subclassifications as well as all available genotype data. Accession numbers for whole genome sequencing data and RNA-seq data should also be given in this table.
B5.
In table 5, targeted association data for 2 variants are given. For the first variant, genotypes for 182 dogs are given, for the second 185 dogs. This is inacceptable, if the study included 264 dogs (according to methods). These numbers must be consistent! If no DNA from a dog is available, then it should not be included in the study. Genotyping success rate for a few specific candidate variants in a few hundred DNA samples should ideally be 100%, but certainly no less than 95%.
B6.
The introduction does not clearly distinguish between somatic variants and germline variants that may lead to TC (lines 37-52).
B7.
Within a dog breed, linkage disequilibrium is typically very long (several Mb). It is therefore essential to perform a detailed haplotype analysis to identify a reliable critical interval. The authors should identify the exact recombination breakpoints that define this critical interval. This might potentially reduce the list of candidate genes that need to be considered.
B8.
Loss-of-function variants in TPO lead to familial hypothyroidism, not to thyroid cancer! The proposed candidate causal variants are not plausible at all. SIFT predictions are highly unreliable and do not prove causality! They may provide a minimal amount of circumstantial evidence at best.
B9.
The TPO variants appear to be associated with TC risk (Table 5). However, the association is not perfect. This strongly suggests that they do not represent the causal variants themselves, but are only in linkage disequilibrium with the true causative variant,
B10.
I would consider a reliable mapping of a TC risk locus in this cohort as substantial scientific advance. Identifying the true causative variant for this risk locus will require additional and better designed experiments than what the authors have presented so far. At the moment, the authors cannot provide any plausible mechanistic hypothesis how the risk factor leads to an increased thyroid cancer risk. TPO loss of function results in hypothyrodidism, not in thyroid cancer. TPO gain of function could potentially lead to an increased production of reactive oxygen species and thus be tumor promoting. However, gain-of-function alleles are typically inherited with a dominant mode of inheritance. The cited "evidence" for an association of TPO to human TC is extremely weak. In my opinion, the cited refs. 42 and 43 do not provide any reliably evidence that TPO variation may be causally related to cancer. I concede that the TPO gene is intriguing to its role in thyroid function. However, without a clear mechanistic hypothesis how TPO variants might predispose to cancer, the authors should deifnitively not prematurely exclude all other genes in the critical interval (and any gene that might be influenced by regulatory elements in the critical interval).
B11.
The entire manuscript and especially the discussion do not sufficiently reflect the existing knowledge on germline genetic risk factors for cancer. The authors are encouraged to seek additional help from an expert on (canine) cancer genetics to bring this manuscript into a publishable form.
Round 2
Reviewer 3 Report
The authors ignored most of my comments. I consider the small revisions to the original manuscript are insufficient. I repeat my main concerns:
(1)
Mapping a risk locus for a familial cancer represents a major scientific advance. However, the authors must genotype more dogs to achieve sufficient power. The selective sweep analysis is inappropriate. The GWAS will most likely become very convincing, if the authors increase their cohort size. This does not take very long and is not overly expensive. Genotyping more cases on SNV arrays will also provide the chance to identfiy further recombinations of the cancer-associated haplotype and to reduce the size of the critical interval.
(2)
The entire part on identifying candidate causative variants is not convincing. The authors still did NOT provide a PLAUSIBLE explanation why loss of function variants in TPO should cause thyroid cancer in Dutch German Longhaired pointers. Loss of function variants in TPO have been PROVEN to cause hypothyroidism in humans and several other dog breeds! The authors speculate about altered levels of H2O2, which is very difficult to measure. However, if these variants indeed affect the function of TPO, then the levels of thyroid hormones would also be expected to be altered. This is easy to measure. Why don't the authors provide any experimental evidence for their claims?
(3) Additional personal remarks:
I speculate that the true causative variant is a non-coding regulatory variant. I therefore don't think that it is a good strategy to only focus on protein-changing variants.
Table 5 indicates that heterozygous dogs have a significantly increased risk for thyroid cancer. Thus, the mode of inheritance is most likely semi-dominant and not strictly recessive.
Author Response
Please see the attachment.

This manuscript is a resubmission of an earlier submission. The following is a list of the peer review reports and author responses from that submission.
Round 1
Reviewer 1 Report
Manuscript ID CJAS-2015-127.R2 entitled « Recessive deleterious mutations in the TPO gene underlying 2 familial thyroid follicular cell carcinomas in Dutch German 3 longhaired pointers» by Yu et al. presents two potential causal mutations, recessively inherited, in the TPO gene in GLP dogs that influence the apparition of thyroid cancer. This manuscript of very good language level shows a very complete analysis of dog’s genome that allowed to find and present very conclusively potential causal mutations for this type of cancer. I congratulate the authors for their very high-quality work. I have only very few comments of minor importance.
